# Modeling geographic vaccination strategies for COVID-19 in Norway

**Louis Yat Hin Chan**[1]*, **Gunnar Rø**[1], **Jørgen Eriksson Midtbø**[1], **Francesco Di Ruscio**[1], **Sara Sofie Viksmoen Watle**[2], **Lene Kristine Juvet**[2], **Jasper Littmann**[3,4], **Preben Aavitsland**[3,5], **Karin Maria Nygård**[6], **Are Stuwitz Berg**[2], **Geir Bukholm**[3,7], **Anja Bråthen Kristoffersen**[1], **Kenth Engø-Monsen**[8], **Solveig Engebretsen**[9], **David Swanson**[10], **Alfonso Diz-Lois Palomares**[1], **Jonas Christoffer Lindstrøm**[1], **Arnoldo Frigessi**[11], **Birgitte Freiesleben de Blasio**[1,11]

**1** Department of Method Development and Analytics, Norwegian Institute of Public Health, Oslo, Norway, **2** Department of Infection Control and Vaccines, Norwegian Institute of Public Health, Oslo, Norway, **3** Division of Infection Control, Norwegian Institute of Public Health, Oslo, Norway, **4** Bergen Centre for Ethics and Priority Setting (BCEPS), University of Bergen, Bergen, Norway, **5** Pandemic Centre, University of Bergen, Bergen, Norway, **6** Department of Infectious Diseases and Preparedness, Norwegian Institute of Public Health, Oslo, Norway, **7** Faculty of Chemistry, Biotechnology and Food Sciences, Norwegian University of Life Sciences, Ås, Norway, **8** Smart Innovation Norway, Halden, Norway, **9** SAMBA, Norwegian Computing Center, Oslo, Norway, **10** Department of Biostatistics, MD Anderson Cancer Center, University of Texas, Houston, Texas, United States of America, **11** Oslo Centre for Biostatistics and Epidemiology, University of Oslo and Oslo University Hospital, Oslo, Norway

* imlouischan@gmail.com

**Data Availability Statement:** The Norwegian health data used in this study is sourced from the Emergency Preparedness Register for COVID-19 (Beredt C19). The right to access one's own personal data in health registries follows from the

## Abstract

Vaccination was a key intervention in controlling the COVID-19 pandemic globally. In early 2021, Norway faced significant regional variations in COVID-19 incidence and prevalence, with large differences in population density, necessitating efficient vaccine allocation to reduce infections and severe outcomes. This study explored alternative vaccination strategies to minimize health outcomes (infections, hospitalizations, ICU admissions, deaths) by varying regions prioritized, extra doses prioritized, and implementation start time. Using two models (individual-based and meta-population), we simulated COVID-19 transmission during the primary vaccination period in Norway, covering the first 7 months of 2021. We investigated alternative strategies to allocate more vaccine doses to regions with a higher force of infection. We also examined the robustness of our results and highlighted potential structural differences between the two models. Our findings suggest that early vaccine prioritization could reduce COVID-19 related health outcomes by 8% to 20% compared to a baseline strategy without geographic prioritization. For minimizing infections, hospitalizations, or ICU admissions, the best strategy was to initially allocate all available vaccine doses to fewer high-risk municipalities, comprising approximately one-fourth of the population. For minimizing deaths, a moderate level of geographic prioritization, with approximately one-third of the population receiving doubled doses, gave the best outcomes by balancing the trade-off between vaccinating younger people in high-risk areas and older people in low-risk areas. The actual strategy implemented in Norway was a two-step moderate level aimed at maintaining the balance and ensuring ethical considerations and public trust. However, it did not offer significant advantages over the baseline strategy without geographic prioritization.

Health Registry Act § 24 and from GDPR chapter III, art. 15 (with certain exceptions in the Personal Data Act §§ 16 and 17). Ethics approval has been obtained for the use of data in this article, authorized through the Norwegian Health Preparedness Act, paragraphs 2-4. More information regarding this approval can be found at https://www.fhi.no/en/id/infectious-diseases/coronavirus/emergency-preparedness-register-for-covid-19/. The aggregated case data and computational code for two models are accessible to the public and shared on GitHub, available at https://github.com/folkehelseinstituttet/COVID19_vaccination-IBM or https://github.com/imlouischan/corona-no, and https://github.com/folkehelseinstituttet/COVID19_vaccination-MPM. However, due to privacy considerations, vaccination data is not accessible to the public. Instead, demonstration of pseudo data with added noise on the original data is used. More information on vaccination data is available at https://www.fhi.no/en/va/norwegian-immunisation-registry-sysvak/ The mobility data was collected and provided by Telenor Norway. Requests for access to mobility data should be directed to Telenor Research via email: TelenorResearch@telenor.com.

**Funding:** This project received funding from the research project "COVID-19 in Norway: A Real-Time Analytical Pipeline for Preparedness, Planning, and Response during the COVID-19 Pandemic in Norway" (Norges forskningsråd grant number 312721) to BFdB and the centre BigInsight (Norges forskningsråd grant number 237718) to AF. The funders played no role in shaping the study design, conducting data collection and analysis, making decisions about publication, or preparing the manuscript.

**Competing interests:** The authors declare that they have no conflict of interest.

Earlier implementation of geographic prioritization could have more effectively addressed the main wave of infections, substantially reducing the national burden of the pandemic.

## Author summary

We utilized two geographic-age-structured models (an individual-based model and a meta-population model) to conduct a scenario-based analysis aimed at evaluating strategies for geographic prioritization of COVID-19 vaccines in Norway. By reconstructing the dynamics of COVID-19 transmission from January to July of 2021, we compared various alternative vaccination strategies through model simulations, given the limited number of vaccine doses. We found that prioritization of vaccines based on geographic location, alongside considering age, was preferable to a baseline strategy without geographic prioritization. We assessed the selection of which municipalities to prioritize and the degree of prioritization they should receive. Our findings indicated that optimal strategies depended on whether the aim was to minimize infections, hospitalizations, ICU admissions, or deaths. Trade-offs in infection growth between municipalities and subsequent risk-class allocations (such as age groups) were the primary factors influencing optimal vaccine allocation. Furthermore, we found that earlier implementation of most geographic prioritization strategies was advantageous in reducing the overall burden of COVID-19.

## 1 Introduction

The COVID-19 pandemic has placed an enormous burden on public health systems globally. Since the first detection of the SARS-CoV-2 virus in Wuhan, China, in late December 2019 [1], it rapidly spread to other countries and triggered a worldwide health crisis [2]. The first imported case in Norway was confirmed on the 26th of February 2020 [3]. On the 12th of March 2020, the Norwegian government implemented a national lockdown that involved closure of schools and non-essential businesses, border controls, travel restrictions and social distancing measures [4–6]. The measures were gradually scaled back, and from the autumn of 2020, the government shifted its policy to regional differentiation based on local transmission levels [7, 8]. Until the start of 2021, the government maintained a strict mitigation strategy to avoid overloading the healthcare system, prevent deaths, and focus on early testing and isolation of cases, along with a locally-based contact tracing policy.

At the turn of 2020/2021, in the face of the emergence of the Alpha variant and limited global and domestic vaccine production and supply, Norway launched a mass COVID-19 vaccination program [9], mainly distributing the two messenger RNA (mRNA) vaccines, Comirnaty (Pfizer-BioNTech) and Spikevax (Moderna), to the population. The government initially prioritized vaccines to protect individuals most at risk for severe illness, vaccinating those aged 65 years and above, along with people with underlying medical conditions [10–12]. Additionally, healthcare workers were given priority due to their high exposure and risk of transmitting the virus to vulnerable patients.

Due to substantial variations in infection levels across the country, with urban areas, notably in and around the capital city of Oslo, experiencing particularly high infection rates, the Norwegian government decided to implement a geographic prioritization of vaccines. Initially, on the 9th of March 2021, a moderate geographic prioritization of vaccines was introduced in

Oslo and several neighboring municipalities [13]. Subsequently, on the 19th of May 2021, a more significant redistribution was implemented, which both increased the amount of the prioritization and broadened the geographic scope of vaccine distribution [14]. By the 31st of July 2021, around 3.6 million (68% of the total population) and 1.8 million (34% of the total population) had received their first and second doses, respectively [15].

Selecting a strategy for distributing a vaccine in limited supply during an ongoing pandemic is a complicated task. Particularly, there is a trade-off between direct protection of vulnerable groups versus indirect protection and overall reduction of transmission. Additionally, effective vaccination strategies must account for local COVID-19 epidemiology. Geographic prioritization of COVID-19 vaccines can be guided by several factors, including infection rates, hospitalization rates, and mortality rates, while also accounting for the ability to control transmission over time without resorting to lockdown or other measures with significant societal consequences and costs. For this reason, the optimal strategy may vary depending on the objective and aim of the vaccination program.

Some modeling studies have explored vaccination strategies, focusing on age, health condition and occupation as the primary targets considered for prioritization [16–27]. Most studies concluded that prioritizing high-risk individuals such as the elderly, those with underlying diseases and healthcare workers is preferred for minimizing deaths. On the other hand, prioritizing younger people who have higher contact rates is better for reducing transmission and hence infections. A few modeling studies have investigated geographic allocation of vaccines taking into account spatial heterogeneities as the secondary consideration on top of age [28–31]. Bertsimas et al. optimized the location of vaccination sites to minimize deaths in the US [28]. Lemaitre et al. and Molla et al. optimized by age and region simultaneously based on optimal control theory and showed that vaccine allocation to high incidence areas is the optimal strategy to reduce deaths in Italy and Finland, respectively [29, 30]. Grauer et al. found that deaths could be significantly reduced if vaccines were distributed focusing on one region at a time, using a computational model with Brownian agents [31].

In this study, we document the modeling approach used by the Norwegian Institute of Public Health to support policymakers with the COVID-19 vaccination strategy in Norway [32–35]. The approach includes an individual-based model (IBM) and a geographic-age-stratified meta-population model (MPM). Our aim is to perform a retrospective evaluation of the early Norwegian vaccination strategy and examine the effectiveness of alternative geographic prioritization strategies. Our mathematical models are calibrated to historical epidemiological and vaccination uptake data to capture the local COVID-19 transmission dynamics. To compare geographic prioritization strategies, we vary three key factors: (i) the number of vaccine doses redistributed, (ii) the number of municipalities prioritized, and (iii) the timing of the start of geographic prioritization. These strategies, along with the actual vaccination strategy, are evaluated relative to a national roll-out without geographic prioritization. To assess the effectiveness of these strategies, we explore separately outcomes of infections, hospitalizations, ICU admissions and deaths by calculating the relative risk reduction (RRR) of cumulative outcomes during the first seven months of 2021. Results of the two models are presented side-by-side to examine the robustness of results and highlight any potential structural biases.

## 2 Materials and methods

### 2.1 Ethics statement

Ethics approval has been obtained for the use of data in this article, authorized through the Norwegian Health Preparedness Act, paragraphs 2–4. More information regarding this

approval can be found at https://www.fhi.no/en/id/infectious-diseases/coronavirus/emergency-preparedness-register-for-covid-19.

## 2.2 Data sources

The health data used in this study, covering the period from the 1st of January to the 31st of July 2021, were provided by the Norwegian emergency preparedness register for COVID-19 (Beredt C19) [36]. The database contains individually merged data collected from national health and other administrative registers. Data including positive tests, importations and deaths were obtained from the Norwegian Surveillance System for Communicable Diseases (MSIS) [37]. Additionally, data on hospitalizations and ICU admissions were collected from the Norwegian Intensive Care and Pandemic Registry (NIPaR) [38, 39]. Finally, the vaccination data were obtained from the Norwegian Immunization Registry (SYSVAK), a national electronic immunization registry that records vaccination status of all individuals [40, 41].

The data were pre-processed and aggregated by date, municipality, age, and risk status. The risk status, or medical risk group, refers to individuals with one or more defined diseases or conditions (e.g. organ transplantation, immunodeficiency, diabetes, etc.) that increase the risk of severe disease and death from COVID-19 [42]. Further details about the data used in this study can be found in S1 Text.

## 2.3 The two geographic-age-structured models

Our study employs two stochastic mathematical models, an individual-based model (IBM) and a meta-population model (MPM), to conduct analyses of COVID-19 vaccine distribution scenarios in Norway. The models were developed to simulate the Norwegian population of approximately 5.4 million individuals, taking into account their demographics in terms of age and geographic location. Both models consider the actual age profile and geographic distribution of the population, and simulate the spread of SARS-CoV-2 from January to July 2021, when the majority of the first doses of vaccines were delivered to the population. The distribution of population size and density across the Norwegian municipalities is illustrated in Fig I in S1 Text. Moreover, we rely on a social contact study survey conducted in 2017 [43] to determine the heterogeneity of contact patterns between individuals of different ages, while the contacts between and within regions are based on mobility data [44].

To capture the transmission dynamics of the virus, we employ a Susceptible-Exposed-Infectious-Removed (SEIR) transmission model that accounts for symptomatic individuals who may require hospitalization and have a risk of mortality due to severe infection. The models also incorporate interventions such as vaccination and restrictions of social contacts. The distribution of vaccines follows the priority order established by the Norwegian government, while the changes in non-pharmaceutical interventions (NPIs) and population contact behavior are captured by the transmission rates, as described in the model calibration section below.

**2.3.1 The individual-based model (IBM).** The individual-based model (IBM), also known as an agent-based model (ABM), captures the complex individual variability and local interactions involved in the transmission dynamics of COVID-19 in Norway. The 356 municipalities in Norway are subdivided into 13,521 cells, and individuals within each cell are assigned ages ranging from 0 to 100 years old. To account for different types of contacts, we construct several contact routes, including community, household, school, university, and workplace.

The community contacts refer to interactions that take place outside of the household, school, university, or workplace, and may include activities such as taking public transportation, grocery shopping, and socializing with friends. These contacts are modeled as random

processes following a negative binomial distribution to represent super-spreading events [45] and taking into consideration the age of potential contacts and physical distance between cells. Relative contact rates among different age groups are taken into account based on the social contact study survey [43]. To reflect real-world mobility patterns, we employ a heavy-tailed distribution function based on mobility data from the largest telecommunication company in Norway, Telenor [44, 46]. This function results in a greater likelihood of short-distance transmission compared to long-distance transmission.

Contacts within specific households, schools, universities, and workplaces are assumed to mix randomly. Individuals are assigned to specific households and establishments based on age-structured census data, with middle-aged individuals more likely to reside with their spouse and children, and the elderly often living in pairs or alone. Younger individuals are assigned to schools, while adults are more likely to be in workplaces. Hence, contacts within households and occupations are assumed age-dependent without using an explicit contact matrix, as in the case of community contacts. The IBM was originally developed to study the transmission dynamics of MRSA in Norway [47] and has been adapted to model COVID-19 transmission, as presented in this paper. A main difference is that the current version does not include hospital settings, and healthcare workers are assumed to have the same contact patterns as other occupational groups.

The code for the IBM is available at https://github.com/folkehelseinstituttet/COVID19_vaccination-IBM or https://github.com/imlouischan/corona-no. More details on the IBM can be found in S1 Text.

**2.3.2 The meta-population model (MPM).** The meta-population model (MPM) implements the same epidemiological model as the IBM, but with sub-populations defined by 9 age groups, 2 risk groups, and a varying number of geographic areas based on the prioritization strategy. Contact between age groups is based on the Norwegian survey data [43], while contacts between and within areas are based on the mobility data from Telenor [44, 46].

The model is implemented in R [48] using the odin [49] and mcstate [50] packages. The MPM impelemetation is available at https://github.com/folkehelseinstituttet/COVID19_vaccination-MPM. More details on the MPM can be found in S1 Text.

## 2.4 SEIR transmission and hospitalization model

We employed an SEIR transmission model, which includes pre-symptomatic, symptomatic and asymptomatic infectious compartments, as illustrated in Fig C in S1 Text. In brief, individuals who have not been exposed to the virus or the vaccine are susceptible ($S$). Upon exposure, susceptible individuals ($S$) enter a latent, non-infectious state ($E_a$ or $E_s$), before becoming infectious. Symptomatic individuals first enter a pre-symptomatic state ($I_p$) before developing symptoms ($I_s$), whereas those with asymptomatic infections ($I_a$) are assumed to recover ($R$). Individuals with a symptomatic infection ($I_s$) may recover ($R$), or develop severe illness requiring hospitalization ($H$). Hospitalization can result in discharge and recovery ($R$), or it can lead to ICU admission ($U$). ICU admissions ($U$) are followed by a second stay in the hospital ward ($H$), before discharge and recovery ($R$). Deaths ($D$) are assumed to occur from the symptomatic state or during hospitalization. The time spent in each compartment follows gamma distributions in the IBM and exponential distributions in the MPM, and the parameters are shown in Table A in S1 Text.

The probabilities of hospitalization and death depend on age and risk status, whereas the probabilities of ICU admissions depend solely on age. We consider that older people have a higher probability of hospitalization compared to younger people, and the risk increases exponentially with age, as reported by another study [51]. Similarly, people belonging to the risk

group for severe illness have a higher probability of hospital admission and death than those who do not. For simplicity, we assume that the probability of ICU admission is independent of risk status. The probabilities of hospitalization, ICU admissions, and death are reported in Tables C, G, and H in S1 Text.

To account for regional differences during the simulation period, we scaled the transmission rate parameter $\beta$ by a scaling factor for each municipality. These scaling factors, or relative reproduction numbers, were assumed to be constant and calculated based on the proportion of cumulative number of reported cases and population size between February 2020 and May 2021 in each municipality, and are provided in Section S2.2 in S1 Text. A map showing the distribution of regional scaling factors is also included in Fig I in S1 Text.

We assume a seasonal effect on transmission across all municipalities, with a relative difference of 50% between the maximum (in winter) and minimum (in summer). The seasonality is calculated based on the mean daily temperature for Norway and illustrated in Fig B in S1 Text.

We consider cases imported from abroad and calculate the daily number of importations using monthly data that are aggregated by age and county, as shown in Fig B in S1 Text.

The simulations begin on the 1st of January 2021 concurrent to the vaccine roll-out. To initialize the number of individuals in each compartment, we rely on the results of a separate model used by the Norwegian Institute of Public Health (NIPH) for situational awareness and forecasting of the COVID-19 pandemic in weekly runs, enabling the monitoring of the pandemic in Norway since March 2020 [44, 52, 53]. The situational awareness model provides the total numbers of all age groups and risk factors per county. Within each county, we assume a uniform distribution across municipalities, age groups, and risk factors. This ensures that the total population count is divided among all subpopulations based on their respective population sizes.

## 2.5 Vaccination model

Based on observational studies during the pandemic, which suggest similar real-world effectiveness, we assume that both vaccines (Comirnaty and Spikevax) are equally effective in our models [54].

We vaccinate individuals who are not infected at the time of vaccination and assume that the vaccines are "leaky", meaning that the vaccine effect of both the first and second doses is imperfect [16]. This implies that vaccinated individuals have a lower risk of infection upon exposure compared to unvaccinated individuals. The vaccine effectiveness (VE) of the two doses is assumed to be a two-step monotonically increasing function over time, as shown in Fig B in S1 Text. For simplicity, the MPM incorporates the leaky scheme by utilizing separate compartments. We assume that people receive their second dose 12 weeks after the first dose. We also assume that the time to reach full effect is 14 days after the second dose vaccination, which is consistent with another study [55]. Given the relatively short simulation period of seven months, we decided not to consider waning effects of vaccine-derived immunity in this study.

We characterize vaccine effectiveness by five types of protection, namely against (i) symptomatic infection, (ii) asymptomatic infection, (iii) hospitalization, (iv) death, and (v) transmissibility. The first two effects against infection represent a reduction in susceptibility to the virus, while the last effect against transmissibility is translated into a reduction in infectiousness for infected people. Vaccine effectiveness parameters are provided in Table B and detailed information with references on the assumptions of vaccine effectiveness can be found in S1 Text.

## 2.6 The actual Norwegian vaccination strategy

The national government established guidelines for prioritization and administration of the vaccine program, while the municipalities were responsible for vaccinating their residents [10].

During the initial roll-out at the turn of 2020/2021, vaccines were allocated to municipalities based on the number of people above 65 years old exclusively. Elderly people including residents in nursing homes and those with underlying medical conditions were given priority to protect those at greatest risk of severe illness and death from SARS-CoV-2 infection [56]. In practice, healthcare workers were also given priority. In our models, we positioned healthcare workers after the elderly population, and the distribution of vaccines to municipalities was based on the size of at-risk populations, with priority given in age-descending order (refer to Table J in S1 Text for more information):

1. Elderly aged above 65 years (divided into three age categories)

2. Healthcare workers

3. Individuals aged 18–64 years with underlying medical conditions (divided into three age categories)

4. Adult population aged above 18 years without underlying medical conditions (divided into four age categories)

The geographic prioritization of COVID-19 vaccines in Norway was carried out in two steps. The initial step involved making a priority ranking of the 356 municipalities based on the cumulative hospitalization rates per resident population. On the 9th of March 2021, around the time most individuals above 85 years old had been vaccinated, the government decided to implement moderate geographic prioritization, whereby 20% additional vaccine doses were allocated to 10 municipalities, including the capital Oslo and four municipalities in the surrounding Viken county that had experienced consistently high infection levels over time [13]. These 10 municipalities, referred to as the *Plus* group, received vaccines at the expense of 330 municipalities, referred to as the *Minus* group, while the remaining 16 municipalities received the same number of vaccine doses and were referred to as the *Neutral* group. Later in March, the municipality distribution key was changed to population proportions of adults above 18 years old. The second phase began on the 19th of May 2021, around the time most individuals above 65 years old had been vaccinated, with a stronger geographic targeting of vaccines, where 60% extra doses were allocated to the prioritized regions [14]. This strategy involved adding 19 municipalities to the *Plus* group, primarily major cities and municipalities in Viken, while 319 and 13 municipalities were placed in the *Minus* and *Neutral* groups, respectively. The prioritization continued until all eligible adult individuals in the *Plus* group aged above 18 years had been offered vaccination.

To simulate the vaccination strategy used in Norway, we used vaccination data recorded by SYSVAK from January to July 2021. Real data from SYSVAK revealed that the prioritization plan was not consistently followed considering vaccine deliveries to each municipality and the vaccination of age groups within the municipalities, possibly due to vaccine hesitancy, logistical challenges, and unforeseen events. In our simulations, we proceeded by extracting the daily count per municipality, age, and risk status of individuals who received their first dose of the vaccine and assumed that they would all receive their second dose 12 weeks later. This assumption was supported by the high uptake rate of 90% among the adult population as of July 2021 (see Table J in S1 Text), as well as the fact that 93% of those who received the first dose also received the second dose [36].

## 2.7 Model calibration

We calibrated our models to fit the national registry data between the 1st of January and the 31st of July 2021. The calibration process involved two steps, with the first step focusing on fitting the trend and age distributions of hospitalizations, and the second step focusing on fitting the age distributions of ICU admissions and deaths.

In March, the sharp increase in hospitalizations due to the third wave of COVID-19 infections and the arrival of the Alpha variant prompted the government to impose restrictions on social gatherings and travel. To account for changes in NPIs, we implemented two change points for the national transmission rate on the 28th of January and 11th of March 2021.

The calibration involved 12 free parameters including transmission rate parameters and susceptibility to infection parameters. The transmission rate parameters $\beta$ controlled the time-varying behavior (contact rate) in three time periods, while the susceptibility to infection parameters described the relative probability of being infected upon exposure in each of the nine 10-year age group. Due to computational costs, the models were calibrated to fit the national daily hospitalization data by time and age, and we used a Latin hypercube sampling (LHS) to explore the parameter space. The goodness-of-fit was evaluated using a least squares method by the IBM, while the MPM used a Poisson likelihood.

The IBM was additionally calibrated to fit the proportions of the four transmission routes based on information provided by general practitioners regarding likely transmissions and collected by Norwegian Surveillance System for Communicable Diseases (MSIS) [37]. The data revealed that 45.1%, 38.6%, 9.9%, and 6.2% of cases from January to July 2021 were infected within households, community, workplaces, and schools, respectively. We used these proportions to establish specific weights for scaling the transmission rate in each setting.

After calibrating the trend of hospitalizations and transmission, we estimated the probabilities of ICU admission given hospitalization and the probabilities of death given infection using age-stratified numbers of infections, hospitalizations, ICU admissions, and deaths over the entire simulation period. The fitted IBM and MPM to epidemiological data are shown in Fig 1. Both models estimate a significantly higher number (on the order of a factor two) of infections than reported cases, particularly in the younger age groups, indicating a high degree of under-ascertainment of cases [57]. Further details about the calibration of the two models can be found in S1 Text.

**2.7.1 Sensitivity analysis on the hospitalization probabilities.** Due to the lack of seroprevalence data in Norway, the precise number of infected individuals remains uncertain. To address the uncertainty surrounding the total number of infected individuals in Norway, we conducted a sensitivity analysis by doubling the probabilities of hospitalization given infection used in our main analysis. As a result of this sensitivity analysis, the model produced a number of infections that closely matched the reported cases. However, it is important to note that there may still be additional unreported cases that were not accounted for [57].

Nonetheless, in our main analysis, we incorporated the expected under-detection of cases by halving the probabilities of hospitalization given infection based on international literature sources [51, 58]. For a more detailed explanation of our assumptions, please refer to S1 Text.

## 2.8 Alternative strategies of geographic prioritization

We developed alternative geographic prioritization strategies for vaccine distribution based on the actual Norwegian vaccination strategy. We ensured the comparability of our models by keeping the calibrated parameters and setting the daily available doses identical across all strategies. As we distributed vaccine doses in pairs (first and second doses), all the numbers of doses refer to the first doses shown in Fig K in S1 Text.

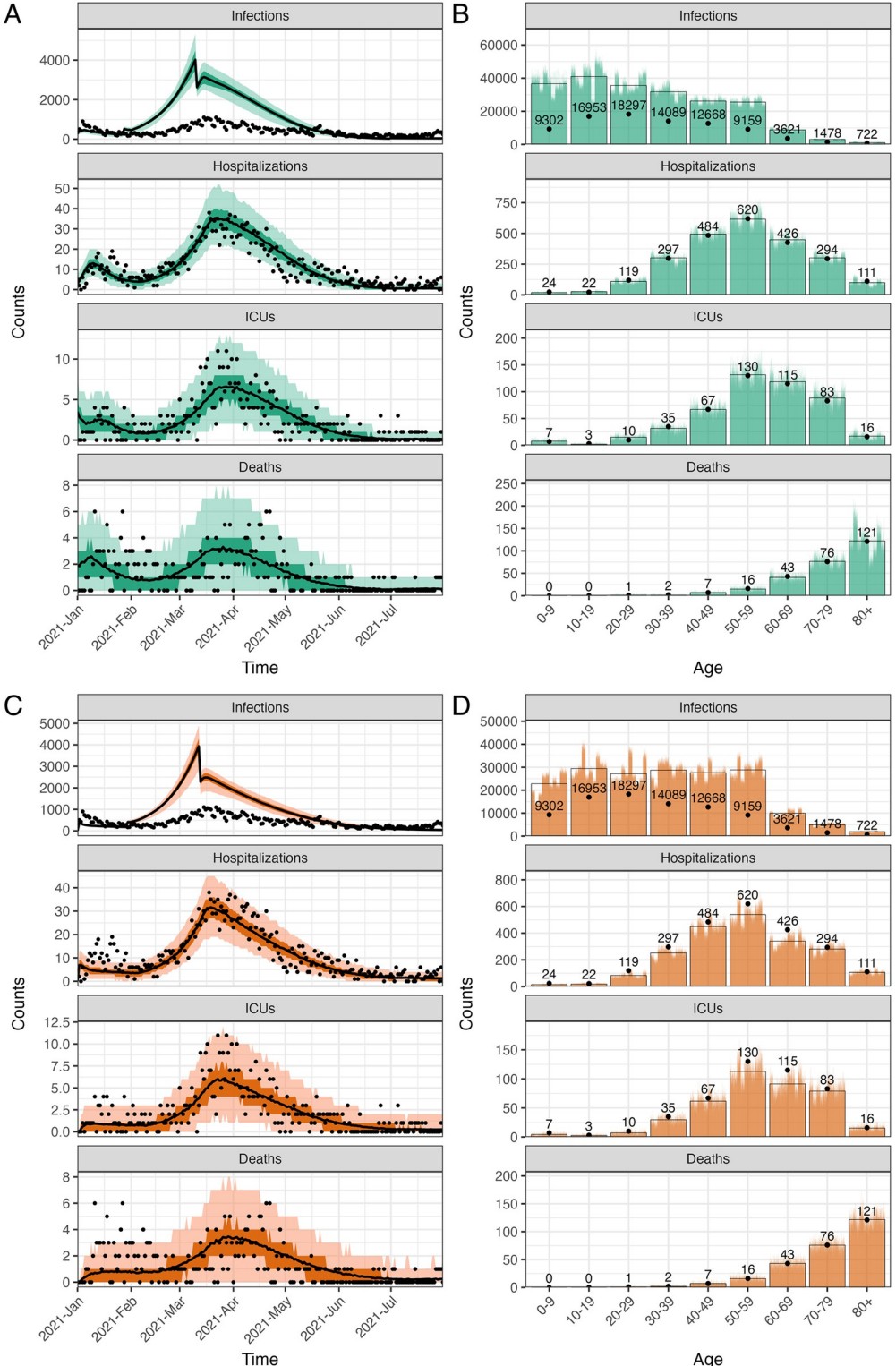

**Fig 1. The calibrated individual-based model (IBM) and meta-population model (MPM) with data in the actual vaccination strategy.** The upper and lower panels show the IBM and MPM, respectively. A and C: The time series data of all ages. The lines show the model fits with their 50 and 95% prediction intervals represented by colored areas based on 1000 simulations. The dots show the observed data. B and D: The age distribution of total counts. Each of the 1000 colored bars shows the counts from one simulation and the full bars with black borders show the mean of all

simulations. The data are shown in dots with their exact numbers. During the whole period, there were 86289 confirmed cases, 2397 hospital admissions, 466 ICU admissions, and 266 deaths registered.

We categorized the 356 municipalities into three groups: *Plus*, *Minus*, and *Neutral*, which corresponded to receiving extra doses, fewer doses, and the same number of doses, respectively. We simplified the geographic prioritization process into a single step and assumed that all municipalities followed the national guidelines for age and risk group prioritization. These strategies were defined by three key parameters:

- $\Delta p$: The proportion of extra vaccine doses provided to the *Plus* group. The proportion received by each municipality is calculated relative to a national baseline strategy without geographic prioritization of eligible adult individuals aged above 18 years. We consider up to a maximum of 300% additional doses, corresponding to the municipalities in the *Plus* group receiving quadruple the number of doses relative to the adult population fraction.

- $\Delta n$: The shift of municipality priority towards the *Plus* group ($\Delta n > 0$) or *Minus* group ($\Delta n < 0$). The selection of municipalities is based on the relative reproduction numbers of the regions. The shifts in either direction lead to a decreased population in the *Neutral* group, and the proportion of the population in the three groups changes accordingly. Geographic maps illustrating the distribution of various alternatives is presented in Fig 2, and more detailed descriptions of the shifts are provided in the following.

- $\Delta t$: The timing of the start of the vaccination program measured in months. The geographic prioritization strategy is assumed to be implemented on the first day of the months, i.e., on the 1st of January, February, March, April, May, June and July 2021.

To derive alternative strategies, we employed the same grouping of municipalities as the original selection implemented in the second step of the actual strategy, with 24, 13, and 319 municipalities in the *Plus*, *Neutral*, and *Minus* groups, respectively. Given the large differences in sizes of municipalities, such as Oslo accounting for 12% of the Norwegian population, while other municipalities may contain only less than a thousand people, it is essential to consider the proportion of the population that is prioritized. In original selection ($\Delta n = 0$), the *Plus*, *Neutral*, and *Minus* municipalities represent approximately 50%, 20% and 30% of the Norwegian population, respectively.

The alternative strategies were selected in the following way:

1. We rank each municipality according to their relative reproduction numbers, as described in Section S2.2 and Fig L in S1 Text. Alternative target areas are chosen by shifting the priority of five municipalities at a time, considering the ranking and their priority group.

2. To increase the prioritized geographic region relative to the adopted strategy ($\Delta n > 0$), we add municipalities to the *Plus* group (originally 24 municipalities) in successive steps. The top five highest ranked municipalities in the *Neutral* and *Minus* groups are moved to the *Plus* group. This procedure is repeated until 40 additional municipalities are shifted to the *Plus* group, i.e., 29, 34, ..., and 64 municipalities are in the *Plus* group.

3. To increase the non-prioritized geographic region relative to the adopted strategy ($\Delta n < 0$), we add municipalities to the *Minus* group (originally 319 municipalities) in successive steps. The bottom five lowest ranked municipalities in the *Plus* and *Neutral* groups are moved to the *Minus* group. This procedure is repeated until 30 additional municipalities

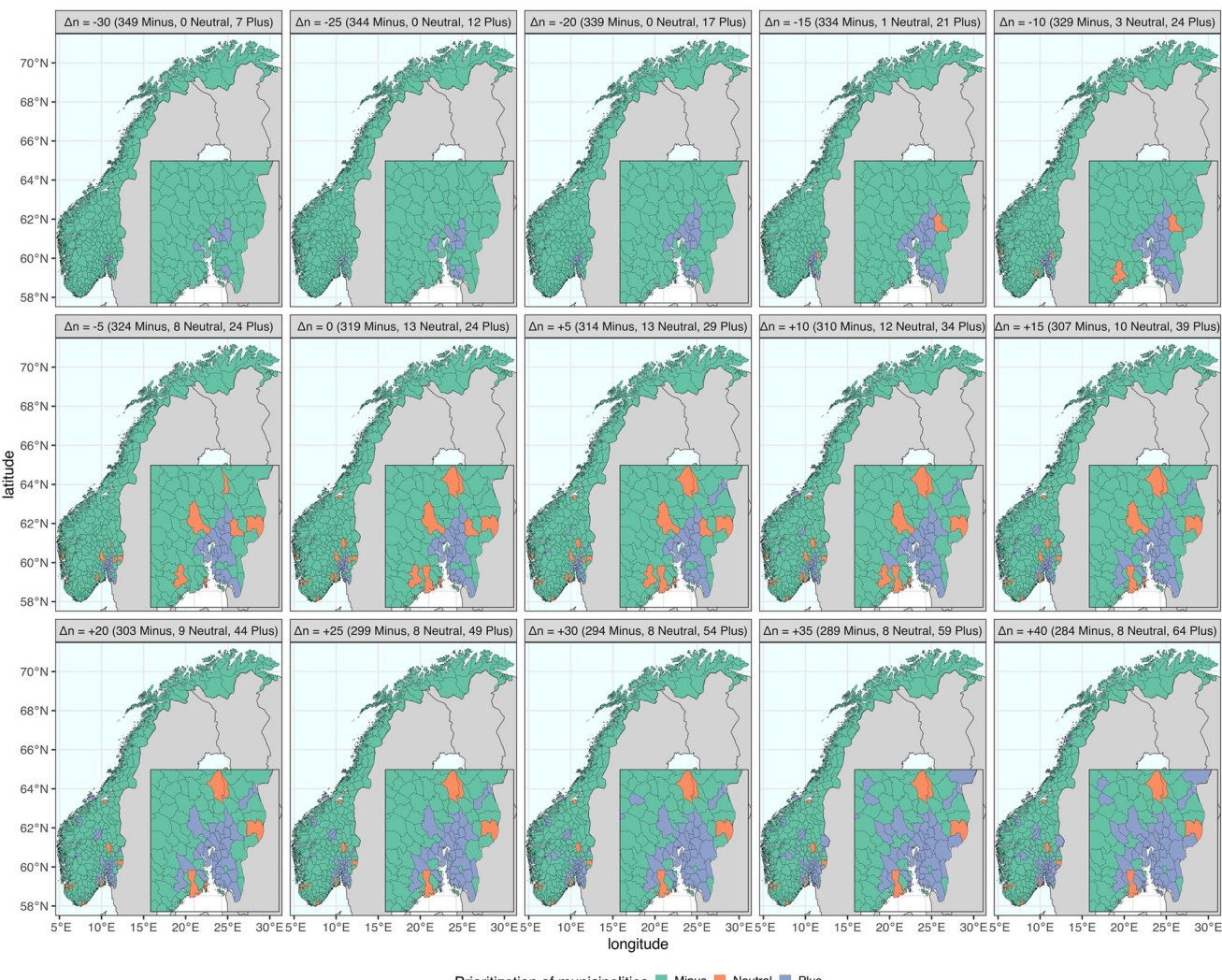

**Fig 2. The geographic distribution of municipality prioritization in alternative strategies.** The 356 municipalities are classified into three groups, represented by the colors green (*Minus*), orange (*Neutral*), and blue (*Plus*). Each strategy comprises a different number of municipalities in each group. The baseline strategy ($\Delta n = 0$) represents the selection made in the real-world implementation. The shifts of municipality priority $\Delta n$ are indicated in brackets, and the number of municipalities in each group is shown accordingly in each panel title. The maps were created using two R packages: "rnaturalearth" for country-level data and "fhidata" for Norwegian municipality-level data. All open-source shape files are licensed under Creative Commons BY 4.0 (CC BY 4.0) and CC0 1.0 (No Copyright), respectively. The country-level data is sourced from Natural Earth (https://www.naturalearthdata.com), while the Norwegian municipality-level data was obtained from Geonorge (https://kartkatalog.geonorge.no/metadata/norske-fylker-og-kommuner-illustrasjonsdata-2020-(klippet-etter-kyst)/7408853f-eb7d-48dd-bb6c-80c7e80f7392).

are shifted to the *Minus* group, i.e., 324, 329, . . ., and 349 municipalities are in the *Minus* group.

4.  For each of the 14 selected geographic regions in Step 2 ($\Delta n > 0$) and Step 3 ($\Delta n < 0$), as well as the baseline strategy ($\Delta n = 0$), we select a value for prioritization of vaccine doses $\Delta p$ ranging from 0% up to a maximum of 300%, and a date of implementation $\Delta t$ from January to July 2021. The maximum value of $\Delta p$ is determined by the population ratio between *Minus* and *Plus*, which is presented in Table I in S1 Text. The number of extra vaccine doses provided to the *Plus* group for each strategy combination of $\Delta n$ and $\Delta p$ is shown in Fig M in S1 Text.

5. As a baseline for comparison of the alternative strategies, we consider a national strategy without geographic prioritization (i.e. $\Delta p = 0$).

To evaluate the strategies, we conducted 1000 simulations in both IBM and MPM. We used 100 stochastic seeds for each of 10 calibrated parameter sets. Each simulation in the IBM took approximately 3 minutes, while each simulation in the MPM took approximately 20 seconds. All simulations were run in parallel on large computer clusters. We compared each iterated simulation of the alternative strategies to the national baseline strategy without geographic prioritization. To assess the reduction in health outcomes, we calculated the relative risk reduction (RRR) using the following formula:

$$\text{RRR} = \frac{x_{\text{Baseline}} - x_{\text{Alternative}}}{x_{\text{Baseline}}}, \tag{1}$$

where $x$ represents the cumulative infections, hospitalizations, ICU admissions or deaths over the entire simulation period. The RRR values indicate the amount by which the alternative strategies reduce the health outcomes compared to the baseline strategy. An RRR = 0 means that the alternative is the same as the baseline, while a positive RRR = 50% means that the alternative strategy reduces the cumulative health outcome by 50% compared to the baseline strategy. The higher RRR indicates the more effective the strategy to reduce the burden, and an RRR = 1 represents a disease-free situation. In contrast, a negative RRR = −100% or −200% means doubling or tripling of the outcomes, respectively, compared to the baseline strategy. The following results show the mean values and their 95% confidence intervals (95%CIs) using $\pm 1.96 \frac{\sigma}{\sqrt{N}}$, where $\sigma$ is standard deviation of RRR and $N = 1000$ is the number of simulations.

In addition, to evaluate the actual strategy as a special case, we generated another national strategy without geographic prioritization, using the real vaccination data. We adjusted the vaccine distribution starting from the implementation of the first step of geographic prioritization (i.e., 9th of March 2021) based on the adult population (aged 18 years or older) in each municipality, while keeping all the data prior to that day. We generated 100 realizations corresponding to the 100 stochastic seeds for each of 10 calibrated parameter sets. We chose to compare the actual strategy with this national strategy instead of the fully controlled baseline strategy above for several reasons. Firstly, healthcare workers were given a high priority in the actual strategy, while we vaccinated healthcare workers after the elderly in the baseline strategy. Secondly, the actual strategy in January and February prioritized based on the population of people aged 65 years or older, while we considered all adults throughout the entire period in the baseline. This resulted in prioritized municipalities receiving more doses than in the actual strategy, primarily due to more younger people living in Oslo and its surrounding areas. Thirdly, both the proposed geographic and age prioritization was not strictly followed, especially during the initial phase, in the actual strategy as we could control in our baseline simulations.

## 3 Results

### 3.1 Evaluating the actual strategy

Table 1 presents the total mean number of infections, hospitalizations, ICU admissions, and deaths from the 1st of January to the 31st of July 2021 under the baseline strategy without geographic prioritization, along with the relative risk reduction (RRR) of the actual strategy and best strategies from both models. The RRR for infections, hospitalizations, ICU admissions, and deaths resulting from the actual strategy were nearly zero, implying that the actual strategy was similar to the baseline without geographic prioritization.

                    

**Table 1. The health outcomes and relative risk reduction (RRR) of different strategies compared to the baseline strategy without geographic prioritization.**

| Model | Outcome | Baseline strategy | Actual strategy | Best strategy | | |
|-------|---------|-------------------|-----------------|---------------|---|---|
| | | Count (95% CI) | RRR, % (95% CI) | RRR, % (95% CI) | $\Delta p$, % | $\Delta n$ (Population, % in *Minus*, *Neutral*, and *Plus*) |
| IBM | Infections, $10^3$ | 242.5 (241.2, 243.8) | -0.1 (-0.5, 0.3) | 16.4 (16.1, 16.7) | 300 | −25 (76.3, 0.0, 23.7) |
| | Hospitalizations | 2688 (2676, 2699) | 0.3 (-0.1, 0.7) | 19.4 (19.0, 19.7) | 250 | −20 (72.7, 0.0, 27.3) |
| | ICUs | 522 (520, 525) | 0.0 (-0.6, 0.5) | 19.2 (18.8, 19.6) | 250 | −20 (72.7, 0.0, 27.3) |
| | Deaths | 235 (234, 237) | -0.7 (-1.3, 0.0) | 8.0 (7.5, 8.6) | 100 | + 10 (46.6, 19.9, 33.4) |
| MPM | Infections, $10^3$ | 195.9 (194.9, 197.0) | -0.2 (-1.1, 0.6) | 16.0 (15.7, 16.3) | 300 | −25 (76.3, 0.0, 23.7) |
| | Hospitalizations | 2188 (2177, 2199) | 0.1 (-0.7, 0.8) | 15.4 (15.0, 15.7) | 300 | −25 (76.3, 0.0, 23.7) |
| | ICUs | 419 (416, 421) | 0.1 (-0.8, 0.9) | 15.6 (15.1, 16.1) | 300 | −25 (76.3, 0.0, 23.7) |
| | Deaths | 240 (239, 241) | 0.2 (-0.5, 0.9) | 5.5 (4.9, 6.2) | 70 | −15 (69.5, 0.4, 30.1) |

The third column shows the total mean number (and their 95% confidence intervals) of infections, hospitalizations, ICU admissions, and deaths in the two models. The mean RRR (and their 95% confidence intervals) of the actual strategy and the best strategies are shown in the next two columns. The best strategies, selected separately for each health outcome, are shown in the last two columns, and were based on the starting time ($\Delta t$) of the 1st of January 2021. The brackets in the last column indicate the population fraction of three groups (*Minus*, *Neutral*, and *Plus*).

## 3.2 Alternative municipality priority

In Fig 3, we present the national RRR from two models for various combinations of proportions of extra doses ($\Delta p$) and shifts in municipality priority ($\Delta n$) for alternative strategies, given the starting time ($\Delta t$) of geographic prioritization on the 1st of January 2021. The leftmost column shows the baseline strategy without geographic prioritization ($\Delta p = 0\%$) does not minimize any health outcome.

To minimize infections, a strong geographic prioritization to a small number of municipalities with high transmission levels is optimal. The best strategy involves reducing the priority by moving 25 municipalities to the *Minus* group ($\Delta n = −25$) and giving 300% extra doses to the *Plus* group ($\Delta p = 300\%$), resulting in the highest mean RRR (and their 95% CIs) of 16.4 (16.1 to 16.7)% and 16.0 (15.7 to 16.3)%, which is equivalent to saving 40,063 (39,190 to 40,936) and 31,563 (30,905 to 32,221) infections in the IBM and MPM, respectively. Similar strategies also minimize hospitalizations and ICU admissions. In the IBM, the optimal strategy involves slightly shifting the priority to $\Delta n = −20$ and $\Delta p = 250\%$, which produces the highest mean RRR (and their 95% CIs) of 19.4 (19.0 to 19.7)% and 19.2 (18.8 to 19.6)%, resulting in saving 524 (514 to 534) hospitalizations and 102 (99 to 104) ICU admissions, respectively. In the MPM, the optimal strategy is $\Delta n = −25$ and $\Delta p = 300\%$, with the mean RRR (and their 95% CIs)) of 15.4 (15.0 to 15.7)% and 15.6 (15.1 to 16.1)% for hospitalizations and ICU admissions, respectively, corresponding to saving 342 (334 to 351) hospitalizations and 67 (65 to 69) ICU admissions.

To minimize deaths, a moderate level of geographic prioritization to a larger number of municipalities is likely optimal, unlike the strategies that minimize infections, hospitalizations, or ICU admissions. In the IBM, increasing the priority by moving 10 municipalities to the *Plus* group ($\Delta n = + 10$) and giving 100% extra doses ($\Delta p = 100\%$) is the optimal strategy, which can avoid 20 (95%CI: 18 to 21) deaths with a mean RRR of 8.0 (95%CI: 7.5 to 8.6)%. In the MPM, the strategy that minimizes deaths is $\Delta n = −15$ and $\Delta p = 70\%$, with a mean RRR of 5.5 (95%CI: 4.9 to 6.2)%, corresponding to avoiding 15 (95%CI: 13 to 16) deaths. However, choosing the prioritization strategy that minimizes infections at $\Delta n = −25$ and $\Delta p = 300\%$ results in a negative effect on deaths with mean RRR of -4.3 (95%CI: -5.0 to -3.6)% and -12.7 (95%CI: -13.4 to

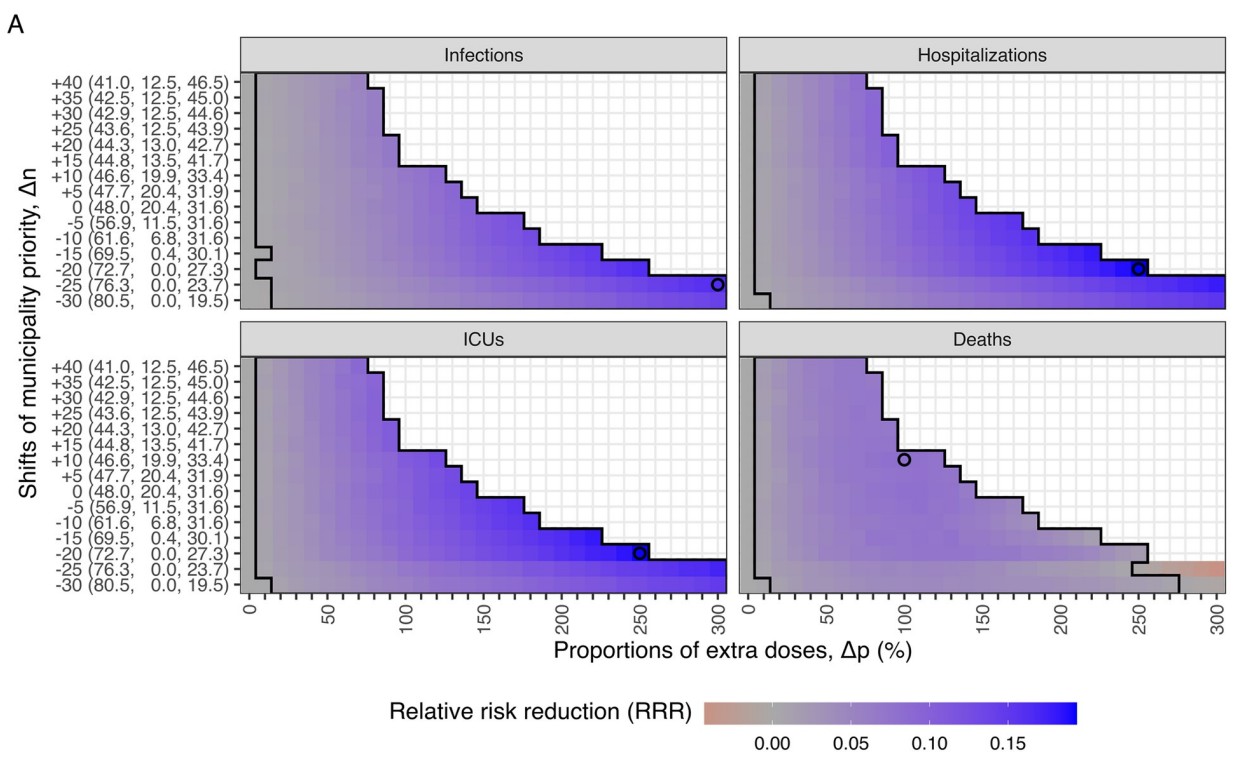

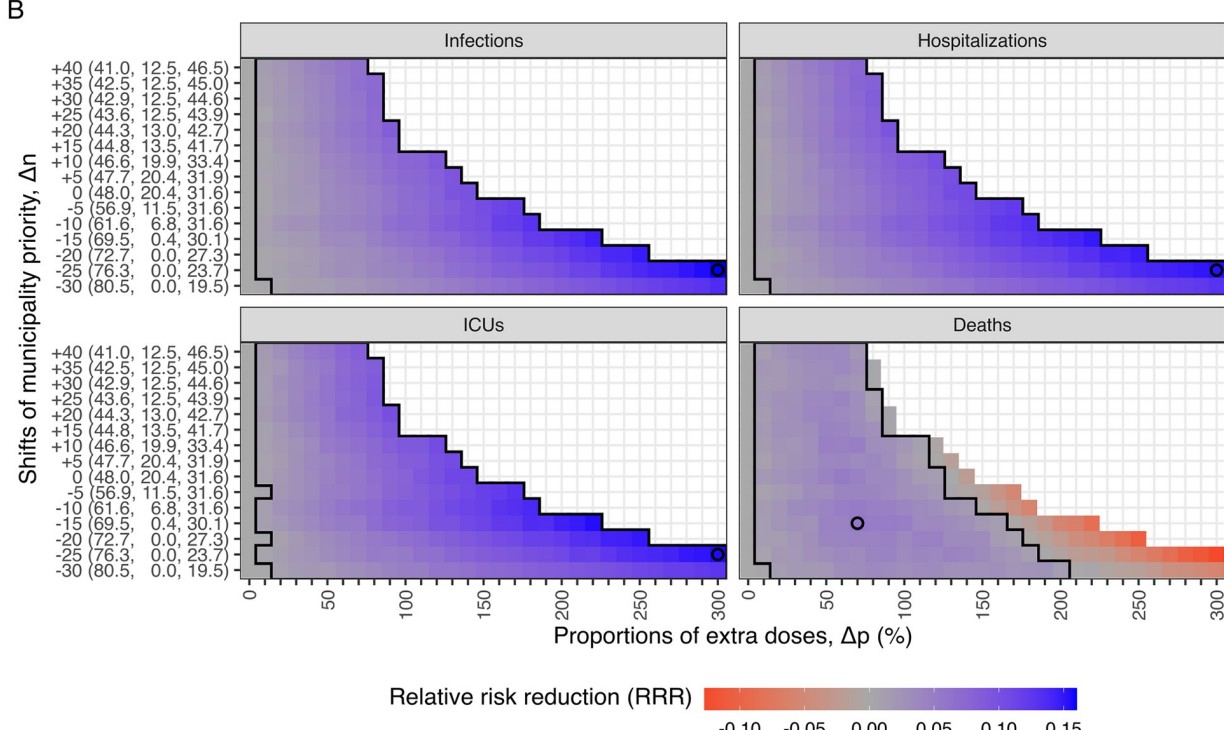

**Fig 3. The mean relative risk reduction (RRR) of health outcomes for the alternative strategies modeled with (A) IBM and (B) MPM.** The mean RRR of infections, hospitalizations, ICU admissions, and deaths is represented by color, ranging from red (negative RRR) to blue (positive RRR). The black empty circles and larger areas illustrate the strategies with the highest mean RRR and positive 95% confidence intervals for each outcome. To minimize infections, hospitalizations, or ICU admissions, the strategies with higher priority levels lead to higher mean RRR. However, for minimizing deaths, the highest level of prioritization has a negative impact, while medium levels of prioritization lead to the highest

mean RRR. The y-axis shows the population fractions (%) of three groups (*Minus*, *Neutral*, and *Plus*) for each shift in municipality priority ($\Delta n$). The geographic distribution of municipality priority ($\Delta n$) can be found in Fig 2.

-12.0)%, leading to 10 (95%CI: 8 to 11) and 29 (95%CI: 28 to 31) extra lives lost compared to the baseline strategy in the IBM and MPM, respectively.

The gains in the *Plus* group are similar across all health outcomes, but the losses in the *Minus* group can be substantial under higher level of prioritization strategies. Fig O in S1 Text shows the trade-off between municipalities, i.e. the *Plus* and *Minus* groups. Similarly, Fig P in S1 Text shows the trade-off between age groups. When more doses are prioritized, the protection of the younger population increases, resulting in a higher average age of infections.

Taking the optimal strategy for minimizing infections in the IBM as an example, Fig 4 shows the trade-off between municipalities by plotting the RRR of each municipality on a map. The municipality-specific RRR of infections, hospitalizations, ICU admissions, and deaths are positive in Oslo and its surroundings because they receive more doses and hence prevent larger local outbreaks. Although the national benefit is larger than the losses in terms of infections, hospitalizations, and ICU admissions, it is not for deaths. Non-prioritized municipalities experience a significant negative impact under higher level of geographic prioritization, creating a large trade-off between age groups. Fig N in S1 Text shows the trade-off by age groups under different geographic prioritization strategies.

### 3.3 Alternative starting time

The starting time of changing from the national to geographic prioritization strategy is also a key factor in the strategy for prioritizing vaccines. We selected a subset of the combinations of $\Delta p$ and $\Delta n$, and explored alternative implementation dates $\Delta t$ ranging from January to July 2021. Our results indicate that earlier implementation yields better control by reducing adverse health outcomes across most prioritization strategies.

Fig 5 shows that the RRR of infections, hospitalizations, and ICU admissions decreases by the starting time across all alternative prioritization strategies. However, the RRR of deaths decreases only when prioritizing in mild and moderate levels (approximately $\Delta p < 200\%$). For higher priority levels (approximately $\Delta p > 200\%$), initiating the geographic prioritization in February or March results in a higher RRR of deaths compared to initiating in January, as shown Figs Q and R in S1 Text. Nevertheless, the moderate level ($\Delta p = 70\%$–$100\%$) with a starting time in January as the previous section shown, the RRR of deaths is the highest among all strategies. Additionally, we found that all the RRR approach zero if the starting time is after May 2021.

### 3.4 Sensitivity analysis on the hospitalization probabilities

Doubling the probabilities of hospitalization does not significantly change the optimal strategies for minimizing infections, hospitalizations, ICU admissions, and deaths. For minimizing infections, hospitalizations, and ICU admissions, the strategies remain similar to those with the highest prioritization level. The moderate level of prioritization is still preferred for minimizing deaths. Similarly, implementing geographic prioritization after January leads to a decrease in RRR, which reaches nearly zero with no benefits if it starts after May 2021. For a more detailed comparison of the two assumptions on hospitalization probabilities, please refer to S1 Text.

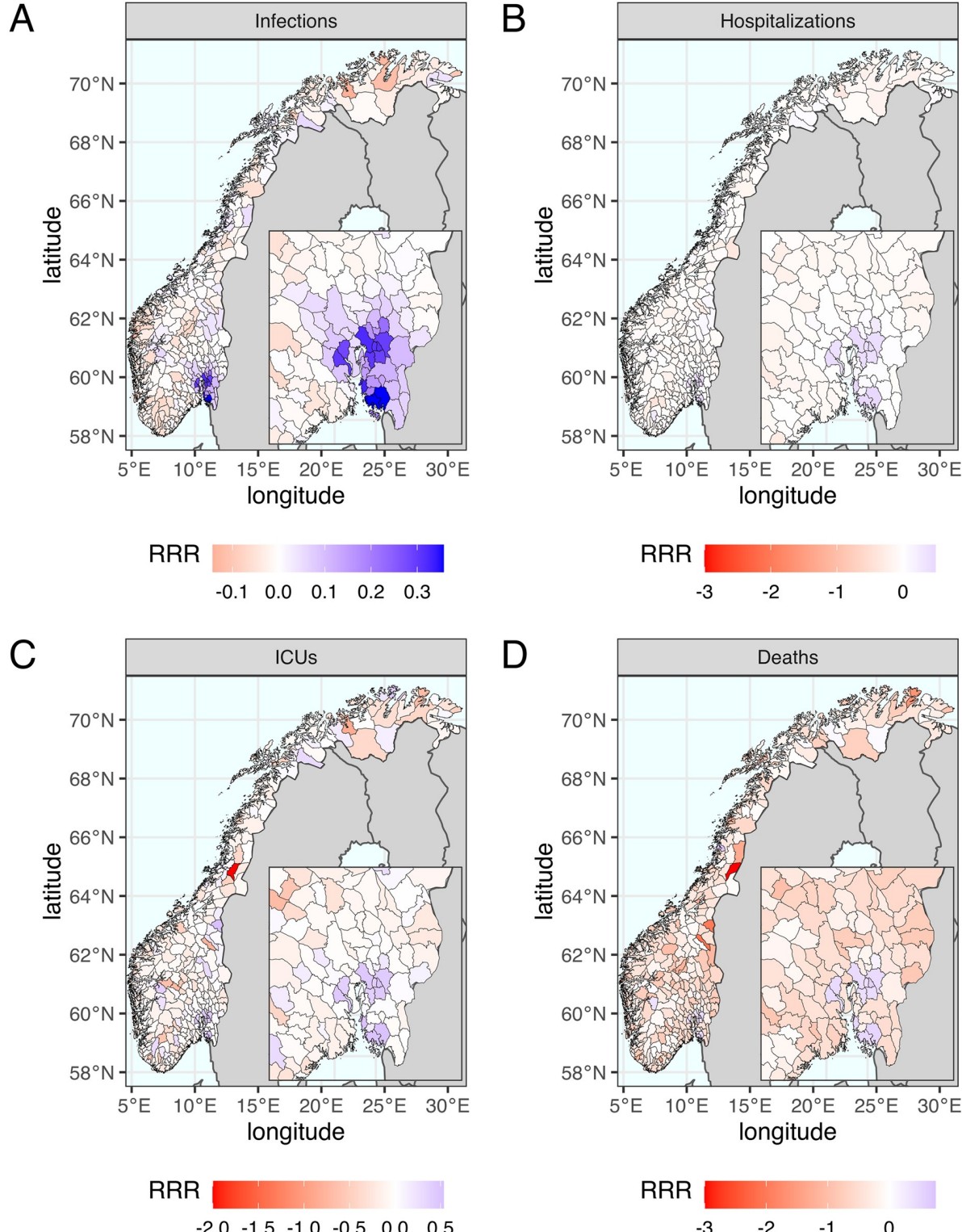

**Fig 4. The geographic distribution of mean relative risk reduction (RRR) of the optimal scenario that minimizes infections.** The optimal scenario is the one of reducing the priority by moving 25 municipalities to *Minus* and giving 300% extra doses ($\Delta n = -25$, $\Delta p = 300\%$). The geographic trade-off between municipalities is illustrated by color from blue to red representing mean values of municipality-specific RRR from positive to negative. The benefits in Oslo and its surroundings are much greater than the drawback in other municipalities to minimize (A) infections, (B) hospitalizations and (C) ICU admissions but not (D) deaths. The maps were created using two

R packages: "rnaturalearth" for country-level data and "fhidata" for Norwegian municipality-level data. All open-source shape files are licensed under Creative Commons BY 4.0 (CC BY 4.0) and CC0 1.0 (No Copyright), respectively. The country-level data is sourced from Natural Earth (https://www.naturalearthdata.com), while the Norwegian municipality-level data was obtained from Geonorge (https://kartkatalog.geonorge.no/metadata/norske-fylker-og-kommuner-illustrasjonsdata-2020-(klippet-etter-kyst)/7408853f-eb7d-48dd-bb6c-80c7e80f7392).

## 3.5 Comparing the two models

Fig 6 presents a comparison between two models regarding the optimal strategy for minimizing infections. Apart from the initial growth being more rapid in the IBM, the primary discrepancy between the two models is the number of infections on the 28th of January and 11th of March 2021, when the transmission rates $\beta$ undergo a change. Specifically, the infection curve from the MPM decreases instantly and more rapidly than that from the IBM. Moreover, the MPM correspondingly exhibits lower trends of hospitalizations and ICU admissions compared to the IBM. However, the MPM shows a higher trend of deaths than the IBM, attributed to the diverse age composition between the two models. More specifically, the IBM shows a greater number of infections among the younger population, whereas the MPM records more deaths in the oldest age group (aged 80 years or older). Furthermore, the complexity of the IBM, which includes heterogeneous contact structures in four transmission routes, contributes to additional variations. For a detailed comparison of the actual vaccination strategy for calibration and the counterfactual scenario without vaccination, please refer to Fig T and Section S5 in S1 Text, respectively.

# 4 Conclusion and discussion

## 4.1 Merit of this study

Early decision-making concerning the allocation of COVID-19 vaccines was critical, given the limited supply of vaccines. Mathematical models provide a powerful tool for investigating and quantifying the impact of different vaccination strategies. However, during the beginning of 2021, inherent uncertainties related to lack of data, the complex dynamics of the COVID-19 pandemic, its uncertain future trajectory, and the lack of knowledge regarding vaccine effectiveness and vaccine supply combined with the need for speedy results made such model-informed assessments challenging [59]. Specifically, one difficulty encountered during the pandemic was the accurate estimation of vaccine effectiveness against transmissibility. The vaccines demonstrated significant efficacy in inhibiting the transmission of the wild-type Wuhan variant. However, this effectiveness diminished with the emergence of each new variant, such as the Alpha variant.

In this study, we conducted a retrospective analysis to evaluate the geographic vaccination strategies for COVID-19 based on the outcome number of infections, hospitalizations, ICU admissions and deaths in Norway. By comparing alternative vaccination scenarios to the implemented strategy and using historical epidemiological data, we propose optimized vaccination strategies that provide valuable insights for guiding policy decisions and enhancing pandemic preparedness in the future.

Our findings demonstrated that early geographic vaccine allocation to areas with high infection levels could reduce COVID-19-related health outcomes. However, the optimal geographic deployment of vaccines, considering the geographic scope and the proportion of redistributed vaccines, depends critically on the specific objective, such as minimizing infections, hospitalizations, ICU admissions, or deaths.

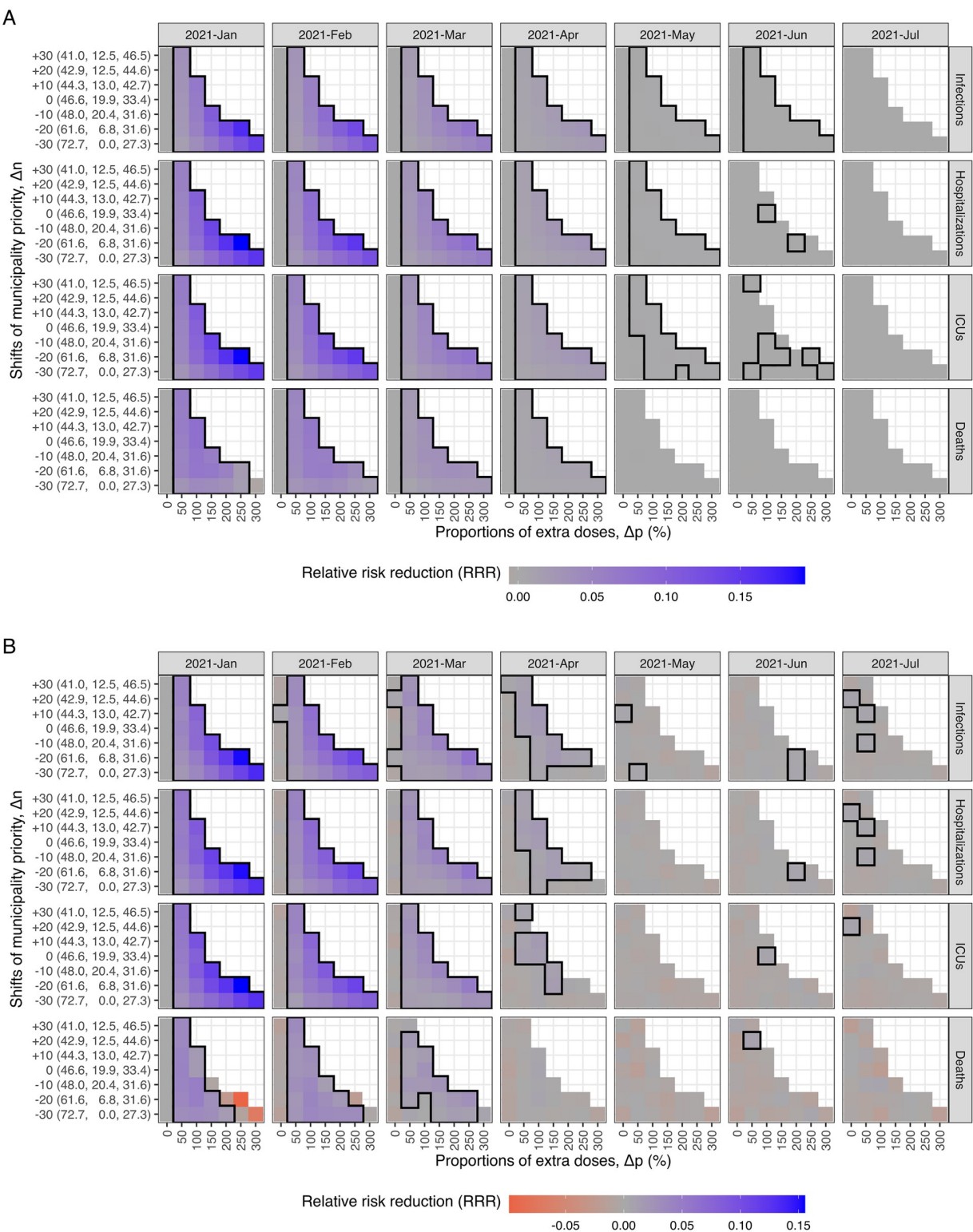

**Fig 5. The mean relative risk reduction (RRR) of health outcomes for the alternative starting time modeled with (A) IBM and (B) MPM.**
The mean RRR of infections, hospitalizations, ICU admissions, and deaths is represented by color, ranging from red (negative RRR) to blue (positive RRR). The panels of columns correspond to alternative start time, with the first column reflecting the results shown in Fig 3. The RRR of infections, hospitalizations, and ICU admissions decrease by starting time, considering all alternative strategies. However, the RRR of deaths decrease only when mild and moderate priority levels (approximately $\Delta p < 200\%$) are implemented. The black square areas indicate positive mean values and their 95% confidence intervals. The benefits are limited if the geographic prioritization started after May 2021.

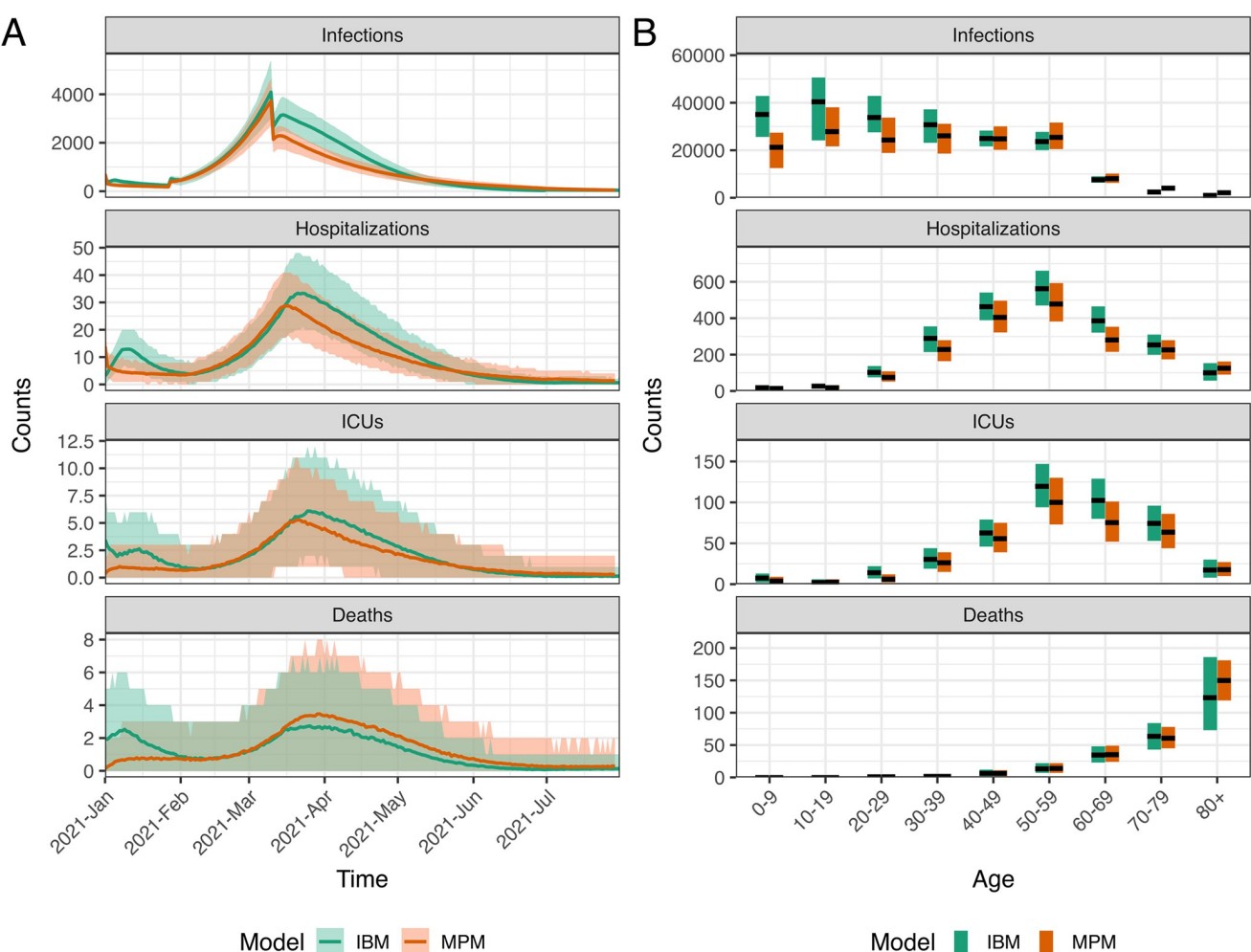

**Fig 6. The comparison of two models for the optimal strategy for minimizing infections.** A: The time series data of all ages. The lines show the model fits with their 95% prediction intervals represented by colored areas based on 1000 simulations. B: The age distribution of total counts. The black lines show the median values, and the colored areas show the 95% prediction intervals based on 1000 simulations.

## 4.2 Infections, hospitalizations, and ICU admissions

For minimizing infections, hospitalizations, or ICU admissions, we found that prioritizing 12–17 municipalities, comprising approximately 25% of the Norwegian population, with the highest infection rates at the beginning of the vaccination campaign, and increasing their vaccine doses by a factor of 3–4, is crucial. This approach ensures earlier vaccination of younger people in those regions while delaying vaccination of older people in the non-prioritized regions, thereby offering direct protection to those at higher risk of infection and indirect protection in high transmission areas, consequently reducing the spread in other parts of the country through internal mobility. This finding is consistent with results by Monod et al. [60], which identified the age group of 20–49 years as the main group sustaining the pandemic, making vaccinating this group in high transmission areas earlier an effective way to control the pandemic. However, it is important to note that targeting a small number of municipalities or a minor proportion of the population can negatively impact mortality and lead to more deaths compared to the baseline strategy without geographic prioritization. This effect is caused by

the significantly higher infection-fatality-ratio among older people and the slower vaccine uptake in that age group within low-priority regions. Notably, this pattern was not observed for hospitalizations because most admissions in Norway occurred in the 50–70 year age group, while the average age of those dying from COVID-19 was above 80 years old [61].

## 4.3 Deaths

To minimize deaths, an early and geographically extended prioritization encompassing 21–34 municipalities comprising approximately 30% of the Norwegian population (alongside approximately 10 *Neutral* municipalities representing approximately 20% of the Norwegian population) and a more moderate level of priority, approximately doubling their vaccine doses, yielded the most favorable outcomes. The optimal strategies slightly differed between the two models, due to variations in population compositions and the geographic breakdown of the country. The IBM modeled all 356 municipalities, one-year age groups and different settings of transmission, while the MPM was limited to a maximum of 15 geographic regions and 9 ten-year age groups. These distinctions also affect the initial conditions and calibration process, resulting in different outcomes in the two models. Due to the exponential increase in death probabilities with age, even small changes in the elderly population led to noticeable differences in the resulting strategies. The results highlight the importance of determining the primary purpose of the vaccination program. In Norway, the government appointed an ethics committee that concluded a vaccination program aimed at minimizing the number of deaths should be chosen [62]. As we demonstrate, this leads to a significantly different optimal strategy than if the objective were to reduce hospitalizations, for instance. Moreover, we anticipate that minimizing disability-adjusted or quality-adjusted life year lost would also necessitate more geographic prioritization, rather than solely focusing on deaths [63].

## 4.4 Consistence with previous studies

Our findings are consistent with previous studies on geographic prioritization [28–31], which also recommended prioritizing high incidence areas. This is due to the higher local transmission rate from a geographic point of view, which is controlled by the regional reproduction numbers in our models. Regions with higher reproduction numbers are the major hubs spreading the disease both internally (within municipalities) and externally (between municipalities). Internally, growth rates of infections are exponential-like expanding given $R_t > 1$ even though vaccines are available. Externally, infectious individuals traveling between regions facilitate the spread of the virus, acting as sources of transmission. We assume that people moving between geographic cells follow the distance function, so short-range movement is more likely to happen. For example, Oslo and its neighboring municipalities form a cluster with high infection rates, and suppressing the growth of infections in the cluster before it becomes unstoppable is notably essential.

Furthermore, the geographic prioritization across populations is associated with age subsequently because of the individual-level prioritization within populations. The trade-off between infections and deaths is mainly due to the age-specific contact and risk as shown in several studies on age prioritization [16–27]. Many of the studies agree that prioritizing the elderly is the optimal strategy to prevent deaths directly, and so as our assumptions in the models to prioritize in a descending age order, starting with those older than middle-aged with higher risk. The selection of strategies between the direct and indirect effect is eventually to protect the older population.

### 4.5 The Norwegian strategy

In Norway, the actual strategy implemented was a moderate level of geographic prioritization, which aimed to maintain a good balance between high-risk and low-risk areas. However, our study found that the actual strategy was similar to the strategy without geographic prioritization, mainly because the second phase of the geographic prioritization as a core part was implemented in May 2021, which was too late to control the pandemic. In fact, this is consistent with our results on starting time.

Our study also highlights the importance of starting geographic prioritization as early as possible in reducing infections, hospitalizations, and ICU admissions. The time switching from the baseline strategy to geographic redistribution could play a significant role during the first three to four months of the implementation. Implementing geographic prioritization after May 2021 was less effective for two primary reasons. First, the main wave of infections had already occurred in March, and the transmission had decreased to $R_t < 1$ as a result of extensive societal lockdown measures. The number of infections in May only reached one-fourth of the peak, making the vaccination strategy at that point less critical. This emphasizes the necessity of geographic prioritization during the first few months to reduce infections and subsequently severe illness. Second, most of the elderly, especially those above 80 years old, had already been vaccinated before May.

Postponing the implementation of geographic prioritization of vaccines reduce the RRR of outcomes on infections, hospitalizations, and ICU admissions. However, when considering a high level of geographic prioritization (such as more than tripling the number of doses to prioritized municipalities), postponing the implementation by one or two months could accelerate vaccination of the oldest age group and directly prevent deaths. Conversely, for mild or moderate levels of geographic prioritization, earlier implementation would be consistently a more effective strategy. This trade-off between priority level and start time is dependent on the decision to minimize either infections, hospitalizations, and ICU admissions, or deaths. Therefore, while earlier implementation of geographic prioritization could prevent more cases, it might not necessarily result in a substantial reduction in deaths. Conversely, postponing the implementation tended to converge towards the national distribution strategy. To make a good balance, there were two potential solutions: (i) implementing an earlier geographic strategy with a moderate level, or (ii) postponing a high priority level of geographic strategy would allow vaccine doses to be used more efficiently in reducing all infections, hospitalizations, ICU admissions, and deaths across the population. However, this presents a difficult ethical consideration during the pandemic [64].

### 4.6 Advantages of using two models

We employed two models in parallel to evaluate the vaccination strategies, and the agreement between the two modeling approaches was essential to validate the results as shown in other studies comparing large-scale computational approaches [65, 66]. The same practice was done in the real-time analysis during the COVID-19 pandemic to determine recommendations of the vaccination strategies in Norway. The national recommendation reports by NIPH are available online [32–35, 67–78]. As an example, the geographic prioritization strategies were evaluated by our two models in the middle of distributing vaccines in February and March 2021 [32–35]. The use of our two models not only assisted in error detection during development but also provided a backup plan for implementation issues, particularly when working under tight deadlines. This retrospective study serves as a valuable tool for preparedness planning, providing valuable insights on how to allocate resources (i.e., vaccines) and how to prepare for future pandemics. In situations with new respiratory viruses where we can assume a

high degree of geographic variation in transmission, it is important to consider geographic prioritization of a limited vaccine supply from the first dose given. This study provides an example of the importance of such prioritization and the value of large-scale computational modeling in public health decision-making.

### 4.7 Limitations

In this study, it is crucial to consider several key assumptions and limitations when interpreting the results. First, we incorporated various simplifications concerning the properties of the virus and its transmission rates throughout the simulation period. We did not account for the growth of the Delta variant from July 2021, resulting in an underestimation of infections during the final phase. However, it is important to acknowledge that the impact of vaccination and its prioritization was beneficial during the Delta variant wave and in the longer term. The effectiveness of the prioritization strategy could have been even greater if potential cases that might have gone undetected were taken into consideration. Furthermore, we acknowledge that Norway, along with several other European countries, implemented a strict societal lockdown, resulting in a significantly lowered transmission rate during the spring. This shutdown of society created an exceptionally unnatural situation. In a scenario without such strong interventions, the effects of vaccine prioritization would likely have been different. Additionally, as our focus was on the short simulation period of 7 months, we did not account for waning vaccine immunity, despite the fact that immunity against infection may decline before the completion of 7 months. Future studies should explore the impact of waning immunity and booster doses given in a longer time frame.

Second, we assumed that all healthcare workers were vaccinated after the elderly (aged 65 or above) in our alternative strategies. This resulted in the elderly aged 65 years or older being vaccinated earlier than in the actual vaccination strategy, leading to an underestimation of the number of deaths in alternative scenarios. This differs from the actual vaccination strategy, which prioritizes front-line essential workers who have a higher risk of preventing outbreaks in hospitals and supporting the healthcare system. Furthermore, we assumed that these healthcare workers had the same contact rates as the general population in our models. In the IBM, we assumed a common contact structure for work-related interactions based on census data. Consequently, we did not distinguish healthcare workers and individuals employed in restaurants or supermarkets from other occupational groups. Incorporating super-spreading events was confined to community transmission, modeled by adherence to a negative binomial distribution [45].

Third, while comparing different strategies, we kept the same transmission rates $\beta$ as calibrated using the actual vaccination strategy. We assumed that the transmission rates vary with a seasonal effect, which is higher in winter and lower in summer [79]. The only difference was the plan of the vaccination program. We assumed that measures such as NPIs would remain exactly the same in all alternatives, although some regions could have large outbreaks. National interventions were captured by the transmission rates at three time intervals, while no regional interventions were considered in this study. We also assumed that the relative reproduction numbers of each municipality were stationary (i.e., constant in time). This made vaccination the only intervention varying in time between regions.

Fourth, our estimates of the regional scale factors for each municipality were based on several simplifying assumptions, potentially impacting the obtained results. Further research is needed to better describe the geographic variation in transmission rates among the municipalities.

Fifth, a key limitation of this study is that the contact matrix used was obtained during the pre-pandemic period. It is likely that the pandemic has changed contact patterns, with older

populations likely reducing their contacts more in the first half of 2021 [80]. Regarding our estimates on the susceptibilities to infection, which decrease by age (except aged 50–59 years), this differs from the findings in other modeling studies [81–83] and cohort studies [84, 85]. Nonetheless, our estimates effectively capture the dynamics of transmission and address the distinction in their contact behavior. The decreasing susceptibilities could be associated with their contact patterns, the age-specific risk ratio of the Alpha variant estimated in Norway [58], or the assumed age-independent vaccine effectiveness. However, all models fitted well to the data and projected more infections than tested cases to reveal under-detection, especially of asymptomatic people or those with mild symptoms.

Finally, we assumed unrealistically that the capacity and logistics of vaccine distribution could always accommodate our scenarios. While the overall logistics in Norway were generally well developed, establishing vaccine centers for preparedness may require additional resources. Two main resources were needed when giving vaccines to the population: the number of doses in stock and human resources needed to vaccinate individuals in the front-line. We fixed the number of (first) doses per day nationally in all scenarios in accordance with the former, but there might be limits in the number of doses for each region, which might realistically cause problems due to the latter. Additionally, there might not be enough staff capacity or time to transport and handle large amounts of doses, although we prioritized some high-risk regions by giving 300% extra doses. Similar limitations exist regarding the assumption of the timing of receiving the second dose to be 12 weeks for everyone. Future studies should address the issue of distributing vaccines with different time delays between two doses as reported by the NIPH [35] and some studies [86–88].

## 4.8 Future considerations and conclusion

To conduct a more comprehensive evaluation of geographic vaccination strategies, it is crucial to consider additional factors [63]. These factors include the impact of lockdown measures, the assessment of both short-term and long-term health outcomes, disease burden estimates that encompass the extended effects of ICU stays and post-infection complications or long COVID, as well as the implications of healthcare prioritization in regions characterized by high infection rates.

During a pandemic, social acceptability plays a crucial role in vaccine distribution. Prioritizing high-incidence areas with a larger allocation of doses can result in smaller municipalities facing inadequate supplies, necessitating a temporary halt to their vaccination programs. The evolving public sentiment surrounding the vaccination policy underscores the potential consequences on public trust and support. Therefore, it is essential to strike a balance between ensuring equitable distribution and maintaining public confidence in vaccination efforts, highlighting the ethical considerations that arise in this context [64].

In conclusion, our study provides important insights into the effectiveness of different COVID-19 vaccine distribution strategies in reducing mortality and morbidity in Norway. Our results suggest that geographic prioritization of vaccines can improve health outcomes during the initial phase of the vaccination program. However, the optimal level of geographic prioritization depends on the specific health outcomes targeted, and a moderate level of prioritization may be optimal for minimizing deaths. It is important to note that our analysis is based on modeling and retrospective data analysis. Employing these methodologies is particularly significant, as they enable the integration of supplementary information that was not available during the initial phase. While the policy of geographic prioritization in Norway has helped reduce the number of deaths, it may not have been optimized to minimize infections or avoid local outbreaks. Nonetheless, our findings can provide valuable guidance for

policymakers in other countries or for future outbreaks, helping them make informed decisions about vaccine distribution strategies.

## Supporting information

**S1 Text. The supplementary document.** This includes more detailed descriptions of the models and additional results.
(PDF)

## Author Contributions

**Conceptualization:** Louis Yat Hin Chan, Gunnar Rø, Jørgen Eriksson Midtbø, Francesco Di Ruscio, Sara Sofie Viksmoen Watle, Lene Kristine Juvet, Jasper Littmann, Preben Aavitsland, Karin Maria Nygård, Are Stuwitz Berg, Geir Bukholm, Solveig Engebretsen, Alfonso Diz-Lois Palomares, Jonas Christoffer Lindstrøm, Birgitte Freiesleben de Blasio.

**Data curation:** Louis Yat Hin Chan, Gunnar Rø, Jørgen Eriksson Midtbø, Francesco Di Ruscio, Anja Bråthen Kristoffersen, Kenth Engø-Monsen.

**Formal analysis:** Louis Yat Hin Chan, Gunnar Rø, Jørgen Eriksson Midtbø, Francesco Di Ruscio, Solveig Engebretsen, Alfonso Diz-Lois Palomares, Jonas Christoffer Lindstrøm.

**Funding acquisition:** Arnoldo Frigessi, Birgitte Freiesleben de Blasio.

**Investigation:** Louis Yat Hin Chan, Gunnar Rø, Jørgen Eriksson Midtbø, Francesco Di Ruscio, Solveig Engebretsen, Alfonso Diz-Lois Palomares, Jonas Christoffer Lindstrøm, Birgitte Freiesleben de Blasio.

**Methodology:** Louis Yat Hin Chan, Gunnar Rø, Jørgen Eriksson Midtbø, Francesco Di Ruscio, Sara Sofie Viksmoen Watle, Lene Kristine Juvet, Solveig Engebretsen, David Swanson, Alfonso Diz-Lois Palomares, Jonas Christoffer Lindstrøm, Birgitte Freiesleben de Blasio.

**Project administration:** Arnoldo Frigessi, Birgitte Freiesleben de Blasio.

**Resources:** Louis Yat Hin Chan, Gunnar Rø, Jørgen Eriksson Midtbø, Francesco Di Ruscio, Sara Sofie Viksmoen Watle, Lene Kristine Juvet, Jasper Littmann, Arnoldo Frigessi, Birgitte Freiesleben de Blasio.

**Software:** Louis Yat Hin Chan, Gunnar Rø, Jørgen Eriksson Midtbø, Francesco Di Ruscio, David Swanson, Alfonso Diz-Lois Palomares, Jonas Christoffer Lindstrøm.

**Supervision:** Preben Aavitsland, Karin Maria Nygård, Are Stuwitz Berg, Geir Bukholm, Birgitte Freiesleben de Blasio.

**Validation:** Louis Yat Hin Chan, Gunnar Rø, Jørgen Eriksson Midtbø, Francesco Di Ruscio, Sara Sofie Viksmoen Watle, Lene Kristine Juvet, Birgitte Freiesleben de Blasio.

**Visualization:** Louis Yat Hin Chan, Gunnar Rø, Jørgen Eriksson Midtbø, Francesco Di Ruscio, Birgitte Freiesleben de Blasio.

**Writing – original draft:** Louis Yat Hin Chan, Gunnar Rø, Jørgen Eriksson Midtbø, Francesco Di Ruscio, Sara Sofie Viksmoen Watle, Lene Kristine Juvet, Jasper Littmann, Preben Aavitsland, Karin Maria Nygård, Are Stuwitz Berg, Geir Bukholm, Anja Bråthen Kristoffersen, Kenth Engø-Monsen, Solveig Engebretsen, David Swanson, Alfonso Diz-Lois Palomares, Jonas Christoffer Lindstrøm, Arnoldo Frigessi, Birgitte Freiesleben de Blasio.

**Writing – review & editing:** Louis Yat Hin Chan, Gunnar Rø, Jørgen Eriksson Midtbø, Francesco Di Ruscio, Sara Sofie Viksmoen Watle, Lene Kristine Juvet, Jasper Littmann, Preben Aavitsland, Karin Maria Nygård, Are Stuwitz Berg, Geir Bukholm, Anja Bråthen Kristoffersen, Kenth Engø-Monsen, Solveig Engebretsen, David Swanson, Alfonso Diz-Lois Palomares, Jonas Christoffer Lindstrøm, Arnoldo Frigessi, Birgitte Freiesleben de Blasio.

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
