## [Decision Letter · Decision Letter 0]

10 Nov 2023

Dear Dr Chan,

Thank you very much for submitting your manuscript "Modeling geographic vaccination strategies for COVID-19 in Norway" for consideration at PLOS Computational Biology. As with all papers reviewed by the journal, your manuscript was reviewed by members of the editorial board and by several independent reviewers. The reviewers appreciated the attention to an important topic. Based on the reviews, we are likely to accept this manuscript for publication, providing that you modify the manuscript according to the review recommendations.

Sincerely,

Claudio José Struchiner, M.D., Sc.D.

Academic Editor

PLOS Computational Biology

Virginia Pitzer

Section Editor

PLOS Computational Biology

Reviewer's Responses to Questions

**Comments to the Authors:**

Reviewer #1: The authors provide a clear and well written account of the modeling efforts conducted by the Norwegian Institute of Public Health to evaluate different allocation strategies for COVID-19 vaccines and guide policy making. Methodology is well grounded, with good integration between modeling efforts and calibration using good quality data, the sources of which are (almost) all clearly documented.

I only have two big concerns regarding the work (outlined in points 3 and 4 below), but I believe they should not impede publication as long as they are properly addressed. Points 1 and 2 outline minor improvements.

1) I commend the authors for the good quality of the text and figures, and my only request in this front is to please use the appropriate command for starting quotations quotations: the first mark is flipped as in line 202 and in the Acknowledgments section; please check if I missed other uses. It also would be nice (but not essential) if Fig 1 followed the color scheme for ABM used in Fig 6, since all the other figures are tastefully colored.

2) The social settings for contacts between agents need to be clarified in two instances:

2.1) The "Community setting" for contacts is not properly defined as it is first mentioned;

2.2) In paragraph between lines 287 and 291, please cite source and give a brief account of the methodology that led to the proportions of infections in each of the settings explored. Moreover, the Community social setting is not listed here and you only mention "other settings"; are these "other settings" the definition for the Community setting that was missing as it was first mentioned?

3) The authors mention that contacts are modeled based on an unpublished social contact studies (105-106).

3.1) Later on, they explain that contact are uniformly distributed for agents within household, schools, universities and workplaces, which leaves only the Community setting (which was already vague) to follow any heterogeneous contact structure. I therefore find the statements in lines (105-106) misleading as it implies data integration of observed contact patterns to a greater degree than actually performed. Again, in lines 506 and 507, the authors mention that heterogeneous contact structure was employed in all four contact settings; this clashes with their explanation that contacts were uniformly distributed in the This needs to be addressed!

3.2) Contact structured is a fundamental element of ABM, and therefore cannot be obscured. Please, address

- if there are valid reasons for this data to remain unpublished;

- if it will be published soon and give enough information for us and the readers to be able to find it later on;

- or include reasonable summary of these studies and their results in the SI (if it was already done and I missed it, please cite the appropriate section of the SI in your original lines).

4) The approach to contact modeling was quite limiting, mainly because

4.1.1) if does not account for heterogeneities in the contact venues employed (different types of workplaces can have very different contact structures with terrible consequences for disease propagation like in accounts of meat processing plants in various countries);

4.1.2) it does not account for multiplicity of roles of a contact venue; for example: where does a restaurant or supermarket place in your venue classification? It is certainly a very relevant social setting and their workers can be effective super spreaders, but at the same time this is not a "workplace" for their customers, so are they not meeting with the workers in your contact model? Then, how could the model hope to represent super-spreading events in these and any other setting? And If so, fundamental mechanism of social interaction which are very important for disease propagation and play a large role in indirect protection conferred by good vaccination policies are not being captured at all and the true potential of ABMs is not being realized.

4.1.3) the assignment of roles to agents (e.g. front line workers) in the ABM model is not described in its appropriated place (section 4.2.1) and only assumed in the Results section. How are these roles assigned, and are any other types of roles assigned to the agents? How do the roles affect contacts?

4.2) I do not hope for a rework of the models to improve contact mechanics, and I believe that your primary intent of documenting the efforts for policy design during that pandemic needs to be respected. However, I would expect improvement in the discussion of the limitations of the contact model employed. These limitations and others you already mentioned are shared by most models and this is a point that the community needs to be better informed.

Despite these concerns, this is a very good work. I find that the project's methodology is very sober and responsible: the hypothesis made are clear, fit reasonably with the designed methodology and provide coherent and valuable guides for policymakers. I congratulate the authors for the work and hope they appreciate the points mentioned in a positive light, recommending this manuscript for publication after these improvements.

Reviewer #2: The authors study the effect of geographic prioritization of vaccination on the reduction of COVID-19 infections and severe outcomes , using two models. Their study provides valuable guidance for policy makers involved in making decisions about vaccination strategies. The paper is well written and the models are described in detail. I only have a few minor comments that need to be addressed before I can recommend the paper for publication.

• Line 392: The following results shows -> … show

• Line 393: 95% confidence intervals for 1000 simulations. Can be only used if the distribution is (close to) symmetric/normal. In case of a skewed distribution, it’s better to report median and IQR. Did you check whether the distribution was symmetric or skewed?

Same question for other parts of the paper where a 95% confidence interval was used.

• S1.1.4. How did you obtain the percentages 40% asymptomatic / 60% pre-symptomatic?

• S1.2.2. of of travelers -> of travelers

Reviewer #3: This paper is interesting since it presents a study of vaccine allocation based on geographic strategies, which is important regarding large countries with heterogeneous densities. Several previous studies during covid-19 times have been carried out concerning prioritisation according to age in relation to different targets (transmission/deaths/etc.) but here the publication of a method to study geographical allocation of vaccine is quite a novelty. The two presented models are quite complex and robust and should be able to well capture the presented elements. The paper is very well written, with the main article containing (almost) all the important information for a wider audience, and a very complete supplementary material describing in details the two models, the assumptions, the methodology and the detailed results. For these reasons, I recommend its publication in PLOS Computational Biology.

Here are some minor comments, sometimes regarding items presented in the supplementary material that could at least be mentioned in the main document for clarity:

- L105: Little additional information should already be given here regarding the origin of the social contract matrix (before the pandemic, 2017 in Norway, only mentioned in the limitations and supplementary materials)

- L284/Fig1caption/etc.: It's is not specified in the main document how the uncertainty is generated, especially concerning estimated parameters. In the supplementary material, there is the mention of selecting the best 10 sets of parameters of a best fit method, which should be mentioned somewhere in the main text. Are the 95% confidence intervals based on those 10 sets (not so logical) ? Is it with additional stochastic realisations ?

- Table 1/Fig3/L558: There is a clear difference concerning optimal strategies with deaths target between the two models, with only a very short comment "likely due to slightly different population compositions" in the discussion section. Might be good to have a broader analysis here. What could be the elements that could be responsible for this difference? calibration? internal model differences? other things?

- Conclusion and Discussion section: This section is one continuous block of text that is far too long. It would be more readable if it were divided into thematic subsections.

- References: There is unnecessary duplication of the full bibliography, hence references in the main article only mentioned in the supplementary material. The main bibliography of the article should be reduced to the items cited.

**Have the authors made all data and (if applicable) computational code underlying the findings in their manuscript fully available?**

Reviewer #1: Yes

Reviewer #2: Yes

Reviewer #3: Yes

PLOS authors have the option to publish the peer review history of their article (what does this mean?). If published, this will include your full peer review and any attached files.

Reviewer #1: No

Reviewer #2: No

Reviewer #3: No

Figure Files:

Data Requirements:

Reproducibility:

References:

---

## [Editor Report · Decision Letter 1]

8 Jan 2024

Dear Dr Chan,

We are pleased to inform you that your manuscript 'Modeling geographic vaccination strategies for COVID-19 in Norway' has been provisionally accepted for publication in PLOS Computational Biology.

Best regards,

Claudio José Struchiner, M.D., Sc.D.

Academic Editor

PLOS Computational Biology

Virginia Pitzer

Section Editor

PLOS Computational Biology

---

## [Editor Report · Acceptance letter]

29 Jan 2024

PCOMPBIOL-D-23-01277R1 

Modeling geographic vaccination strategies for COVID-19 in Norway

Dear Dr Chan,

I am pleased to inform you that your manuscript has been formally accepted for publication in PLOS Computational Biology. Your manuscript is now with our production department and you will be notified of the publication date in due course.

With kind regards,

Anita Estes
